# A Case of Pulmonary Nocardiosis Presenting with Multiple Cavitary Nodules in a Patient with Thrombocytopenia

Fei Wang [1], Qing Yu [1], Jinye Zhu [2] and Chengli Que [1,*]

1 Department of Respiratory and Critical Care Medicine, Peking University First Hospital, 8 Xishiku Street, Beijing 100034, China; wangfei950904@126.com (F.W.); yu_qing818@126.com (Q.Y.)
2 Department of Hematology, Peking University First Hospital, 8 Xishiku Street, Beijing 100034, China; m13502180710_1@163.com
* Correspondence: quechengli@bjmu.edu.cn; Tel.: +86-8357-5677

**Abstract:** Nocardiosis is an infrequent opportunistic infection that deserves more attention because of its increasing morbidity and high rate of misdiagnosis. Clinical and radiological manifestations as well as antibiograms of pulmonary nocardiosis are diverse. Herein, we report a patient with idiopathic thrombocytopenia treated with glucocorticoids, which later presented with severe pulmonary infection with widely distributed nodules on chest computed tomography. Fungal infection or tuberculosis was under initial suspicion and microbiological tests of bronchoalveolar lavage fluid eventually yielded an affirmative result of *Nocardia cyriacigeorgica*. The patient responded in the beginning; however, therapeutic strategies had to be altered several times due to adverse events. The patient eventually ended up with radiographic resolution at the end of six months. We wish to share some experience in dealing with this disease especially recognizing pulmonary nocardiosis in computed tomography imaging.

**Keywords:** nocardiosis; immunocompromised; radiological features; antimicrobial treatment



## 1. Introduction

Nocardiosis is an uncommon Gram-positive bacterial infection caused by a variety of species of the genus *Nocardia*, which is an omnipresent group of environmental bacteria that may be found in soil, decaying vegetation and other organic substance, as well as in natural waters [1,2]. *Nocardia* usually appear as filamentous Gram-positive branching rods and may be differentiated from *Actinomyces* by weakly acid-fast staining, as *Nocardia* contain mycolic acid in their cell walls [3,4]. *Nocardia* are considered opportunistic pathogens in most situations, mainly causing infections in immunocompromised patients [5]. The causes of immunity defects include solid organ or hematopoietic cell transplantation, recent use of glucocorticoids or immunosuppressant, human immunodeficiency virus (HIV) infection, malignancy, diabetes mellitus and so on. Chronic lung diseases with structural changes are additional risk factors for pulmonary nocardiosis [6,7].

*Nocardia* have the ability to disseminate to virtually any organ, of which lungs are the most frequently involved. Formerly, the *N. asteroides* complex was reported to be the most common species in pulmonary infection. In recent years, with the increasing incidence of this disease and the improvement of species identification technologies, the most reported species varies in different countries and regions [8]. A retrospective research study in 2019 indicated that *N. farcinica* was the most common species in China's mainland, followed by *N. cyriacigeorgica* [9]. Herein, we report a case of pulmonary nocardiosis caused by *N. cyriacigeorgica*, which showed striking imaging manifestations, and improved remarkably after appropriate treatment.

## 2. Case Presentation Section

A 55-year-old male patient presented with complaints of productive cough and dyspnea for one week. His symptoms preluded by a cold without fever and deteriorated gradually, while sputum turned black and viscous. The patient suffered from thrombocytopenia for the past three years with the lowest platelet count of $8000/mm^3$ (normal range: $125,000–350,000/mm^3$) and was under oral glucocorticoids treatment for six months and tapered down to 4 mg of methylprednisolone per day at presentation. Previous laboratory tests including bone marrow aspiration and biopsy failed to identify the etiology. The patient also reported fatigue for a month, especially when climbing the stairs, which was considered as an adverse effect of oral glucocorticoids by a physician of another hospital. He was a lifelong nonsmoker, had no chronic pulmonary disease, diabetes mellitus, any kind of surgery or recent trauma and worked as a community police officer. Chest X-ray at a local hospital showed bilateral multiple nodules. Laboratory examination at our emergency room showed a pH of 7.52 (normal range: 7.35–7.45), a $Pa_{CO2}$ of 31 mmHg (normal range: 35–45 mmHg) and a $Pa_{O2}$ of 51 mmHg on room air in arterial blood gas analysis, a total leukocyte count of $11,200/mm^3$ (normal range: $3500–9500/mm^3$) with neutrophil 91%, a platelet count of $43,000/mm^3$ and an elevated C reactive protein (CRP) of 154 mg/L (normal range: <8 mg/L) in a routine blood test. The patient was given intravenous moxifloxacin and voriconazole for 4 days in the emergency room along with oxygen therapy and experienced no significant improvement. Then, he was admitted to respiratory ward.

On examination, his body temperature was 36.8 °C, blood pressure 132/80 mmHg, pulse rate 78 per minute and respiratory rate 18 per minute. His pulse oxygen saturation was 95% on oxygen flow of 5 L/min via nasal cannula at rest. Scattered skin purpura on the abdomen and back were noted. Air entry was reduced but no rales were heard on both lungs. Examination of the other systems was unremarkable. Other than the tests mentioned above, the parameters of his hepatic and renal function were almost normal and serum procalcitonin was 0.27 ng/mL (normal range: <0.05 ng/mL). The T-SPOT.TB assay was negative. Plasma (1,3)-beta-D-glucan, galactomannan, serum *Aspergillus* IgG antibody and *Cryptococcus neoformans* antigen were all negative. The peripheral blood $CD4^+$ T cell count was 67 cells/μL (normal range: 404–1612 cells/μL). The level of serum neuro-specific enolase (NSE) increased to three times the upper normal limit (56.95 ng/mL, normal range: <16.3 ng/mL).

Chest computed tomography (CT) imaging on admission revealed bilateral multiple lung nodules of various sizes, in which some had cavities with thick or irregular walls (Figure 1). Given his critical state of immunosuppression, bronchoscopy was performed, which demonstrated mild mucosal swelling of the dorsal segment bronchus of the left lower lobe, with other segmental bronchi smooth and unobstructed. Bronchoalveolar lavage fluid (BALF) was sent to a variety of microbiological tests, among which routine smears for acid-fast bacilli, fungi and other bacteria, as well as culture for fungi and mycobacterium tuberculosis produced negative results successively. Considering the urgent need for pathogenic information, a metagenomic next-generation sequencing (mNGS) of BALF was ordered as well, and eventually identified 116 sequence reads corresponding to *Nocardia cyriacigeorgica*. Meanwhile, nucleic acids of another three kinds of pathogens were detected (Table 1). *Pneumocystis jirovecii* was not considered pathogenically meaningful on account of its poor abundance and not conforming to the patient's radiological characteristics and would be covered by sulfonamide. Aspergillosis could not be entirely excluded because the antifungal therapy provided before bronchoscopy was likely to interfere with the result of mNGS.

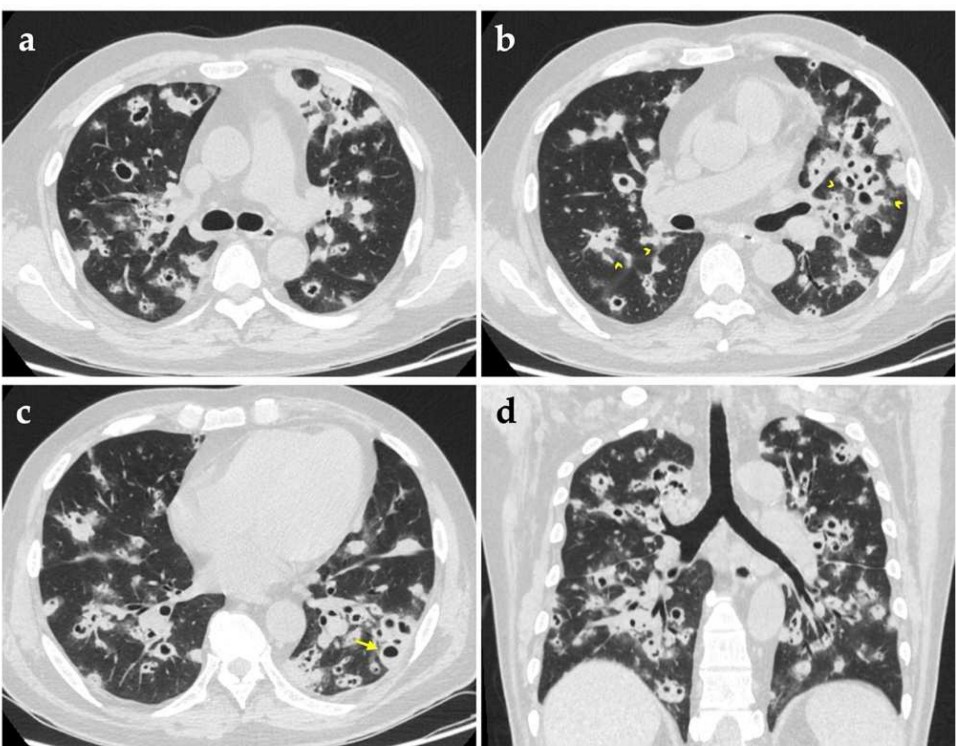

**Figure 1.** Computed tomography images on admission. Axial (**a**–**c**) and coronal (**d**) high-resolution CT images show bilateral multiple nodules of different sizes with cavities. Some are surrounded by ground-glass opacities (arrowheads) and a few have air-fluid level (arrow). (**d**) Shows the peribronchovascular predominant distribution of nodules more clearly.

**Table 1.** Microbial sequences detected in mNGS of the BALF.

| Species | Genus | Reads Counts |
| --- | --- | --- |
| *Nocardia cyriacigeorgica* | *Nocardia* | 116 |
| *Pneumocystis jirovecii* | *Pneumocystis* | 4 |
| *Aspergillus fumigatus* | *Aspergillus* | 1 |
| *Human beta herpesvirus 7* | *Roseolovirus* | 1 |

mNGS: metagenomic next-generation sequencing; BALF: bronchoalveolar lavage fluid.

Empiric anti-infective therapy was modified after admission, consisting of intravenous voriconazole and imipenem-cilastatin. Oral trimethoprim-sulfamethoxazole (TMP-SMX) 320 mg/1600 mg was given every 8 h after mNGS reporting. Voriconazole was continued for 2 weeks. Five days after bronchoscopy, BALF grew *Nocardia cyriacigeorgica* in accordance with the result of mNGS, which was sensitive to imipenem and linezolid, intermediate to amoxicillin-clavulanate and resistant to ceftriaxone and ciprofloxacin. Susceptibility of TMP-SMX was not evaluated due to the corresponding E-test strips being out of stock. The patient's symptoms including dyspnea alleviated upon effective therapy. However, his leukocyte count decreased to 1800/mm$^3$ in the third week of hospitalization, which was attributed to TMP-SMX and/or imipenem-cilastatin. We shifted antibiotics to intravenous amikacin plus oral minocycline according to his antibiogram. His leukocyte count recovered to 2900/mm$^3$ after the adjustment, and his Pa$_{O_2}$ on room air rose to 96 mmHg prior to discharge. The platelet count was stabilized during the hospital stay and was 33,000/mm$^3$ at discharge. The figure describes the two biomarkers which fluctuated with antimicrobial treatment during hospitalization (Figure 2). Thereafter, amikacin and minocycline were continued for a short period, but were abandoned due to intolerance to their adverse reactions. The attempt at oral doxycycline also failed. We finally chose oral amoxicillin-clavulanate as the maintenance therapy.

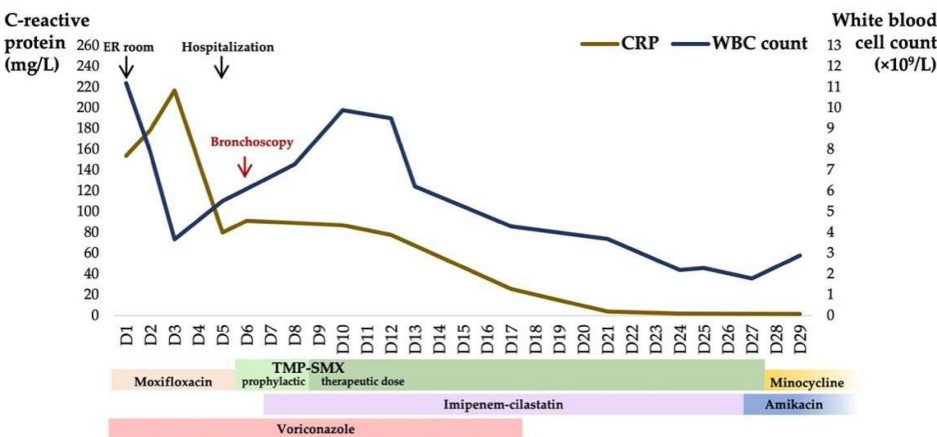

**Figure 2.** Changes in white blood cell count, C-reactive protein level and antimicrobial regimen during hospitalization. Arrows mark several important events.

Three months after discharge, repeated chest CT in the follow-up visit showed that the solid nodules shrunk significantly and most of the cavities vanished. Half a year after discharge, another CT scan showed only several solid micronodules and some irregular linear opacities, without any cavities (Figure 3). Maintenance therapy with amoxicillin-clavulanate was then discontinued.

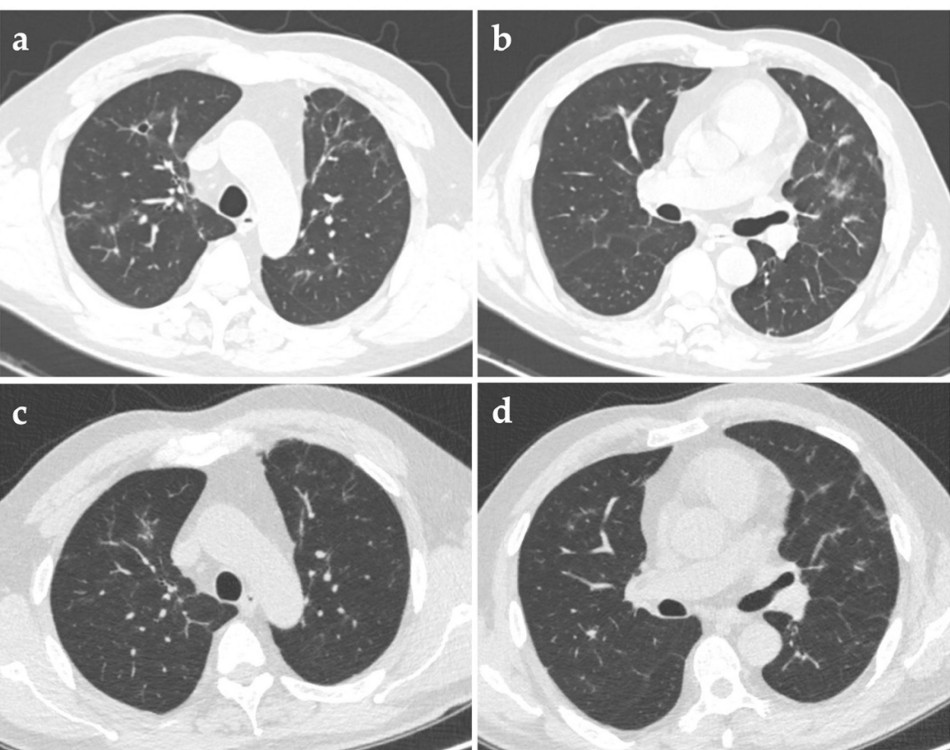

**Figure 3.** Repeated computed tomography (CT) images three months (**a**,**b**) and six months (**c**,**d**) after discharge, showing a significant reduction in both nodules and cavities.

## 3. Discussion

Pulmonary nocardiosis is an uncommon infection caused by *Nocardia* spp., a genus of acid-fast Gram-positive bacilli. *Nocardia* usually result in suppurative infections in immunocompromised hosts, and so it is subsumed into opportunistic pathogens. However, it is worth noting that approximately one-third of nocardiosis patients are immunocompetent [1,10]. In a series of 43 cases of pulmonary nocardiosis, glucocorticoid use was the most common risk factor and accounted for 51.2% of all [11]. The patient in the present

case was treated with long-term glucocorticoids for his thrombocytopenia of unknown origin which was diagnosed as myelodysplastic syndrome in subsequent medical contacts, and thus became the initial factor of this serious illness. His extremely low count of $CD4^+$ T lymphocytes represented the collapse of cell-mediated immunity.

Symptoms of pulmonary nocardiosis are usually nonspecific. Cough with or without expectoration, fever, chest pain, fatigue and weight loss are common and hard to differentiate with other diseases [12]. Radiographic manifestations are important clues for distinguishing pulmonary nocardiosis from other types of pneumonia, tuberculosis or carcinoma. An early study of a few patients indicated that multiple air-space consolidation was the major finding on CT images [13], while a recent study incorporating 25 patients suggested that multiple nodules were the most common imaging manifestation among both immunocompetent and immunocompromised patients [14]. Presentation of single or multiple nodules often mimics malignancy and is difficult to differentiate relying on images alone. Symptoms and correlating laboratory abnormalities with particular clinical history provide crucial hints for diagnosis [15]. In this patient's imaging, ground-glass opacities around those glaring solid nodules, also known as "halo sign", may easily be missed but is worth our attention. The halo sign was first described in patients with invasive pulmonary aspergillosis (IPA), usually caused by hemorrhage or inflammatory cell infiltration surrounding the central area of necrosis pathologically. It was found in cases of aspergillosis most frequently, and has also been reported in infections of *Mucor*, *Candida* spp., *Cryptococcus*, herpes simplex virus, cytomegalovirus, *Mycobacterium avium* complex, *Coxiella burnetti* and so on [16]. Except for the halo sign, nodules in IPA patients often present a classic air-crescent sign when they develop cavitation and may behave as an important distinguishing feature. Therefore, imaging of our patient failed to favor diagnosis of aspergillosis. Tuberculosis and nocardiosis also have resemblances in CT appearance, and both are more susceptible among immunocompromised hosts, while the absence of lymphadenopathy supported nocardiosis over tuberculosis in our case. It should be noted that *Nocardia* exhibit varying degrees of acid fastness due to the constituent mycolic acid inside their cell walls, hence further identification of mycobacterial infection and nocardiosis is quite necessary when patients have positive sputum acid-fast bacilli smears. A small study has indicated that the positive rate of conventional Kinyoun acid-fast staining was 57% (13/23) among *Nocardia* species while that of modified acid-fast staining (using 0.5% sulfuric acid as decolorizer) was 91% (21/23) [17]. Extending time of decolorization may help differentiate *Nocardia* from *Mycobacteria*, as the former turn negative. Modified acid-fast staining using 1% or 0.5% sulfuric acid for decolorization is mostly recommended for identifying the *Nocardia* species [1,17,18].

Standard recommendations for antimicrobial drugs in nocardiosis are not available yet because of the lack of prospective clinical trials. TMP-SMX has been used most widely and has proved effective against most species of *Nocardia* [19]. Myelosuppression as a usual adverse reaction of high-dose TMP-SMX therapy limits its practical application in patients with hematologic disorders. Meanwhile, the susceptibility to antibiotics of different *Nocardia* species varies, and for instance, TMP-SMX is not effective against *N. otitidiscaviarum* according to a spate of reports [20,21]. *N. cyriacigeorgica* is characterized by resistance to ampicillin, amoxicillin-clavulanate and ciprofloxacin, and usually susceptible to TMP-SMX, ceftriaxone, imipenem, amikacin and linezolid [21,22]. Of the tetracyclines, minocycline also shows good activity against most species of *Nocardia* [19]. For critically ill patients, an initial regimen composed of two or three potentially susceptible agents is recommended by mainstream views. Therefore, our patient was given TMP-SMX accompanied with imipenem which successfully controlled the infection at first, but they were subsequently replaced by amikacin and minocycline due to progressive myelosuppression. Minocycline/doxycycline in combination with amoxicillin/clavulanate acid, or the former alone, made an impressive recovery of his general status and chest radiograph, which may partially contribute to his improved immune condition after stopping glucocorticoids, unlike those with persistent immunosuppressive disorders.

## 4. Conclusions

Pulmonary nocardiosis is an uncommon disease which could present as multiple solid nodules with cavities mimicking tuberculosis, aspergillosis or metastatic tumors. Early identification of pathogens by mNGS, as well as individualized antibacterial therapy may improve the prognosis.

**Author Contributions:** Conceptualization, F.W. and C.Q.; investigation, F.W.; clinical management of the patient, Q.Y., J.Z. and C.Q.; writing—original draft preparation, F.W.; writing—review and editing, Q.Y. and C.Q.; supervision, C.Q. All authors have read and agreed to the published version of the manuscript.

**Funding:** This research received no external funding.

**Institutional Review Board Statement:** The study was conducted in accordance with the Declaration of Helsinki. Ethics approval is not applicable.

**Informed Consent Statement:** Written informed consent has been obtained from the patient to publish this paper.

**Data Availability Statement:** All data underlying this article are available in this article.

**Acknowledgments:** The authors thank all staff members seeing and taking care of the patient at Peking University First Hospital.

**Conflicts of Interest:** The authors declare no conflict of interest.

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
