# Peer review of "A Case of Pulmonary Nocardiosis Presenting with Multiple Cavitary Nodules in a Patient with Thrombocytopenia"

_reports, doi:10.3390/reports5020019_

Round 1

Reviewer 1 Report

  1. Line 47 and 55 and on: Normal values of different parameters should be shown
  2. Why was the neuro-specific enolase measured?
  3. Line 99: The results of the antimicrobial susceptibility stated do not include TMP-SMX
  4. Line 109: What adverse reactions did develop?
  5. Line 110: For how long did you treat him in total, and for how long was he put on amoxicillin-clavulanate for maintenance?
  6. Please add some lines at the end summarizing the whole paper as a conclusion
  7. Please state if you had sent cultures for tuberculosis and acid-fast staining

Author Response

Point 1: Line 47 and 55 and on: Normal values of different parameters should be shown.

Response 1: We thank the reviewer for this suggestion. The normal values of complete blood count, C reactive protein, procalcitonin, peripheral blood CD4+ T lymphocyte count and neuro-specific enolase have been added to the text. Normal values of arterial blood gas analysis, except PaO2, are shown. But PaO2, could be varied with age, and fraction of oxygen inhaled.

Point 2: Why was the neuro-specific enolase measured?

Response 2: Neuro-specific enolase (NSE) is generally considered to be associated with small cell lung cancer. Based on the patient’s imaging characteristics, we considered tumors should be ruled out and we tested a series of tumor markers including neuro-specific enolase (NSE), carcinoembryonic antigen (CEA), squamous cell carcinoma antigen (SCC), pro-gastrin-releasing peptide (ProGRP), alpha fetoprotein (AFP) and so on, among which NSE was the only one elevated.

Point 3: Line 99: The results of the antimicrobial susceptibility stated do not include TMP-SMX.

Response 3: The E-test method was used in susceptibility testing in this case. It was a pity that we ran out of stock of TMP-SMX strips at that time. We have explained this in the revised manuscript (line 123).

Point 4: What adverse reactions did develop?

Response 4: The patient was plagued by hair loss and persistent gastrointestinal adverse reactions after administration of amikacin and minocycline. Moreover, intravenous administration of amikacin was very inconvenient in the outpatient setting. We might try in the future the nebulization of amikacin.

Point 5: Line 110: For how long did you treat him in total, and for how long was he put on amoxicillin-clavulanate for maintenance?

Response 5: We treated this patient for six months in total, and administration of oral amoxicillin-clavulanate lasted nearly five months. This information was added to the revised manuscript (line 135).

Point 6: Please add some lines at the end summarizing the whole paper as a conclusion.

Response 6: We have added conclusion part to the manuscript as suggested.

Point 7: Please state if you had sent cultures for tuberculosis and acid-fast staining.

Response 7: We had sent the microbiological tests the reviewer mentioned and related statements were given in the text (line 94-96). 

Reviewer 2 Report

This is an extremely interesting case report presenting a patient with pulmonary nocardiosis. The case is well written and the results well founded. This is a nice example of how metagenomics (NGS) could be applied in clinical practice to solve difficult problems. Nocardia in general is a bacterial species which is not easily cultured, isolated and identified with classic microbiological techniques.  The authors could make a comment whether they had any results from the microbiological lab with classic techniques to compare with the results of the NGS sequencing, as well as results of antibiotic resistance testing. It is well known that  there are Nocardia strains with increased resistance to trimethoprim-sulmamethoxazole. Otherwise, I have not any other specific comments. 

Author Response

Response: We thank the reviewer for the constructive comments. In this case, culture of  bronchoalveolar lavage fluid (BALF) grew Nocardia cyriacigeorgica five days after bronchoscopy in accordance with the result of mNGS, while mNGS saved four days. It was a pity that susceptibility to TMP-SMX failed to be evaluated because its strip for E-test ran out of stock during COVID-19 pandemic at that time (we have explained this in the revised manuscript). According to a recent literature review (Clin Microbiol Infect. 2021 Apr;27(4):550-558), most Nocardia species including N.cyriacigeorgica were still susceptible to TMP-SMX, while a small portion of N.farcinica showed resistance to TMP-SMX in vitro.

Reviewer 3 Report

Abstract (and in the main text):

…broncho-alveolar lavage (BAL)…

…The patient responded in the beginning, however, therapeutic strategies had to be altered..

Instead of „He” use „The patient

GeneraL: please explain all abbreviations on first mention

Main text

 „Gram-positive” (please correct throughout the manuscript)

…may be found in the soil,

…may be differentiated…

contains mycolic acid in its cell wall

L27: Please consider including the following reference:

https://pubmed.ncbi.nlm.nih.gov/32218154/

immunosupressans (e.g.,?)

L38: please provide some examples for changes in the epidemiology based on geographic context

Herein, we report…

L44: male patient

L55: not Emergency Department (ED)?

Instead of „He” use „The patient

L67: what do you mean by „marginally normal” please rephrase

swelling of the dorsal segment bronchus

Aspergillosis could not be entirely excluded

Which society’s breakpoints were used to determine susceptibilities?

at oral doxycycline

L123: Gram-positive bacilli

L147: Candida spp.

Author Response

Point 1: Spelling and grammar suggestions: (1) …broncho-alveolar lavage (BAL)…; (2) …The patient responded in the beginning, however, therapeutic strategies had to be altered…; (3) Instead of „He” use „The patient; (4) “Gram-positive” (please correct throughout the manuscript); (5) …may be found in the soil; (6) …may be differentiated…; (7) contains mycolic acid in its cell wall; (8) Herein, we report…; (9) L44: male patient; (10) L55: not Emergency Department (ED)? Instead of “He” use “The patient”; (11) swelling of the dorsal segment bronchus; (12) Aspergillosis could not be entirely excluded; (13) at oral doxycycline; (14) L123: Gram-positive bacilli; (15) L147: Candida spp. ...

Response 1: We thank the reviewer for pointing out these spelling or grammar mistakes. We have fixed these errors as suggested in the revised manuscript.

Point 2: General: please explain all abbreviations on first mention.

Response 2: We have given clear indications of all abbreviations on first mention as suggested.

Point 3: Please consider including the following reference: https://pubmed.ncbi.nlm.nih.gov/32218154/.

Response 3: We have cited this reference (line 28). Thanks for the friendly recommendation.

Point 4: Please provide some examples for changes in the epidemiology based on geographic context.

Response 4: The reference 8 (line 38) summarized the reported incidence of several countries. According to a report by the U.S. Center for Disease Control and Prevention, 500-1000 new cases occurred in the United States every year from 1972 to 1974, which was equivalent to about 0.23-0.47/100,000. In Canada, which is also located in North America, the incidence rate was about 0.33/100,000 in 1998, and increased to 0.87/100,000 in 2008. The incidence has increased 2.6 times in 10 years. The data could be found in the cited reference, and for making the article concise we didn’t enumerate them.

Point 5: What do you mean by “marginally normal”? Please rephrase.

Response 5: The patient had slightly elevated gamma-glutamyltransferase (γ-GT), while other parameters of renal and liver function were within normal limits. We have changed the word “marginally” to “almost” (line 82).

Point 6: Which society’s breakpoints were used to determine susceptibilities?

Response 6: We used E-test strips for antibiotic susceptibility testing. The breakpoints referred to clinical and laboratory standards institute (CLSI) guidelines. 

Round 2

Reviewer 1 Report

The manuscript has been improved now.

Author Response

Thank you for taking time out of your busy schedule to review our manuscript. Your valuable and constructive comments are very helpful for revising and improving our paper. 

Thanks again for your hard work! 

Best wishes.